# `DivIL`: Unveiling and Addressing Over-Invariance for Out-of-Distribution Generalization

**Jiaqi Wang**[*]                                                                         *jqwang23@cse.cuhk.edu.hk*
*The Chinese University of Hong Kong*

**Yuhang Zhou** [*]                                                                      *ralph.yh.zhou@gmail.com*
**Zhixiong Zhang**[*]                                                                    *zxzhang0216@gmail.com*
**Qiguang Chen**                                                                         *qgchen@ir.hit.edu.cn*
*Harbin Institute of Technology*

**Yongqiang Chen**                                                                       *yqchen24@gmail.com*
*CMU CLeaR Group*

**James Cheng**                                                                          *jcheng@cse.cuhk.edu.hk*
*The Chinese University of Hong Kong*

**Reviewed on OpenReview:** *https://openreview.net/forum?id=2Zan4ATYsh*

## Abstract

Out-of-distribution generalization is a common problem that expects the model to perform well in the different distributions even far from the train data. A popular approach to addressing this issue is invariant learning (IL), in which the model is compiled to focus on invariant features instead of spurious features by adding strong constraints during training. However, there are some potential pitfalls of strong invariant constraints. Due to the limited number of diverse environments and over-regularization in the feature space, it may lead to a loss of important details in the invariant features while alleviating the spurious correlations, namely the *over-invariance*, which can also degrade the generalization performance. We theoretically define the over-invariance and observe that this issue occurs in various classic IL methods. To alleviate this issue, we propose a simple approach Diverse Invariant Learning (`DivIL`) by adding the unsupervised contrastive learning and the random masking mechanism compensatory for the invariant constraints, which can be applied to various IL methods. Furthermore, we conduct experiments across multiple modalities across 12 datasets and 6 classic models, verifying our over-invariance insight and the effectiveness of our `DivIL` framework. Our code is available in `https://github.com/kokolerk/DivIL`.

## 1 Introduction

Modern machine learning methods have achieved great success across various domains such as natural language processing, computer vision, and graph neural networks (Kipf & Welling, 2017; Devlin et al., 2019; Xu et al., 2019). However, these methods heavily rely on the assumption that training and testing data come from the independent and identically distribution(I.I.Dassumption) (Liu et al., 2023; Peters et al., 2016c). When faced with out-of-distribution (OOD) data, almost all of these methods generalize poorly since they are prone to inherit data biases from the train set as shortcuts (Koh et al., 2021; Gulrajani & Lopez-Paz, 2021; Gui et al., 2022; Ji et al., 2022; Xie et al., 2024).

A canonical method for the OOD generalization is invariant learning (IL) based on the invariant principle from causality (Arjovsky et al., 2019; Ahuja et al., 2021; Peters et al., 2016b; Krueger et al., 2021; Yao et al.,

---

[*]Equal contribution.

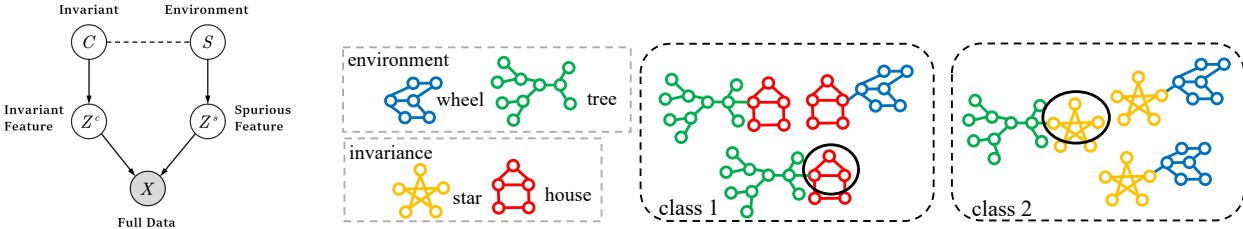

(a) Structural causal model.     (b) Illustration of the over-invariance issue in the graph field.

Figure 1: (a) shows the structural causal model of invariant and spurious features in relation to the invariance and the environments. (b) shows the over-invariance issue in the graph field. Each graph $G$ consists of the invariant subgraph $G_c$ (star, house) and the spurious subgraph $G_s$ (wheel, tree). Previous IL methods alleviate spurious subgraphs while sacrificing important details of invariant subgraphs (The circle is the invariant subgraph $\hat{G}_c$ identified by the model.), causing the over-invariance issue.

2024; Chen et al., 2024; Wang et al., 2024; Xu et al., 2025). As seen in Figure 1a, the basic assumption of IL is that each data is determined by the invariant feature $Z^c$ and the spurious feature $Z^s$ and only learning the invariant features can achieve the success of OOD generalization. Specially, the two variables are unobservable and the invariant one is stable across environments ($Z^c \perp S|C$) while the spurious one changes with environments (S). The key challenge of IL is how to learn the invariant features while alleviating the spurious features. To achieve this goal, various IL methods add regularization to the original Empirical Risk Minimization (ERM) loss, for example, IRMv1, VREx, and Fishr (Krueger et al., 2021; Rame et al., 2022) introduce gradients-induced losses and EIIL, EILLS, and CIGA (Creager et al., 2021b; Fan et al., 2023; Chen et al., 2022) adapt the environment-induced penalties. Others (Peters et al., 2016b; Ahuja et al., 2021; Wu et al., 2022; Sagawa* et al., 2020) apply complex invariant strategies during the training process to extract invariance.

However, the rigorous invariance definition (Arjovsky et al., 2019) must (1) be Bayesian optimal across all environments and (2) completely abandon the spurious feature, which gives a strong restriction to the representation learning. Despite improvement in performance on the test set, most IL methods perform poorly compared to ERM on the train set (Creager et al., 2021b; Kamath et al., 2021a; Krueger et al., 2021). Furthermore, extracting invariant features requires the train data from different environments which are often artificially divided or typically absent in real-world scenarios. Some studies (Lin et al., 2022; Kamath et al., 2021b) have proved that in cases with insufficient environments in the train set, IL fails to distinguish the invariance and the spurious correlation. This reveals two critical dilemmas of IL to capture the invariance: while beneficial for OOD generalization, its over-regularization limits the representation and requires an infinite number of diverse environments.

In this paper, we highlight that during the pursuit of invariance, current IL methods tend to use fewer features to avoid any risk of violating invariance, referred to as the *over-invariance*. Figure 1b demonstrates an example of over-invariance in the graph field where IL predicts the label only by the small subgraph of the invariant subgraph $G_c$ and ignores other part of the graph. However, this may come at the cost of losing enough details and diversity of the invariant feature despite alleviating the spurious correlation, which also degrades the out-of-distribution generalization. Furthermore, we rigorously define the over-invariance and conduct simulation experiments on two classic IL methods, IRM (Arjovsky et al., 2019) and VREx (Krueger et al., 2021), verifying the existence of the over-invariance.

Built upon our observation, we propose a simple and novel method Diverse Invariant Learning (`DivIL`) with a focus on promoting richer and more diverse invariance. Since the quality of the invariant feature plays an essential role in IL, we consider striking a balance between the strong regularizers to alleviate spurious correlations and the detailed capture of the invariant features. We combine invariant penalties and unsupervised contrastive learning (UCL) with random data augmentation to extract domain-wise and sample-wise features, eliminating the reliance on the environments. Meanwhile, we mask the front part of the UCL feature as zero to reduce overfitting to spurious shortcuts (Jing et al., 2021). We evaluate `DivIL` on

an extensive set of 12 benchmark datasets across natural language, computer vision, and graph domains with various distribution shifts, including a challenging setting from AI-aided drug discovery (Ji et al., 2022). We demonstrate that `DivIL` can significantly enhance the performance of invariant learning methods, thereby reinforcing our insight of the over-invariance issue in invariant learning. Our main contributions are:

- We discover and theoretically define the over-invariance phenomenon, *i.e.,* the loss of important details in invariance when alleviating the spurious features, which exists in almost all of the previous IL methods.

- We propose Diverse Invariant Learning (`DivIL`), combining both invariant constraints and unsupervised contrastive learning with randomly masking mechanism to promote richer and more diverse invariance.

- Experiments conducted on 12 benchmarks, 4 different invariant learning methods across 3 modalities (graphs, vision, and natural language) demonstrate that `DivIL` effectively enhances the out-of-distribution generalization performance, verifying the over-invariance insight.

## 2 Background

In this work, we focus on the OOD generalization in the classification task. Specifically, given a set of datasets $\mathcal{D} = \{\mathcal{D}^s\}_s$ collected from multiple environments $\epsilon_{all}$, samples $(X_i^s, Y_i^s) \in \mathcal{D}^s$ are considered as drawn independently from an identical distribution $\mathcal{P}^s$. A model $f = w \circ \Phi$ generically has a representation function $\Phi : X \to H$ that learns the meaningful feature $Z$ for each data and a predictor $w : H \to Y$ to predict the label $\hat{Y}$ based on the feature $Z$. The goal of OOD generalization is to train the model on the train set $\mathcal{D}^{tr} = \{D^s\}_{s \in \epsilon_{tr} \subseteq \epsilon_{all}}$ that generalizes well to all (unseen) environments.

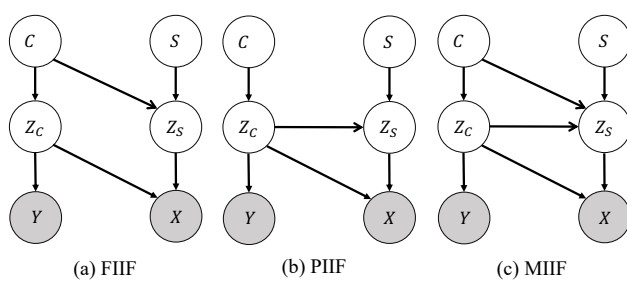

(a) FIIF      (b) PIIF      (c) MIIF

Figure 2: Illustrations of three structural causal models (SCMs).

It is known that OOD generalization is impossible without assumptions on the environments $\epsilon_{all}$. Thus we formulate the data generation process with structural causal model and latent variable model Pearl (2009). The generation of the observed data $X$ and labels $Y$ are controlled by a set of latent causal variable $C$ and spurious variable $S$ as suggested in Figure 1a, i.e.,

$$Z_c := g_{gen}^c(C); \ Z_s := g_{gen}^s(S); \ Z := (Z_c, Z_s)$$

$$X^s := g_{gen}^z(Z); \ Y := f(Z_c).$$

$Z_c$ is the invariant feature determined by the causal variable $C$, $Z_s$ varies with the environment $S$, and label $Y$ is determined by the casual variable $C$. Besides, based on the latent interaction among $C$, $S$ and $Y$, SCM can be further categorized into *Full Informative Invariant Features* (*FIIF*) and *Partially Informative Invariant Features* (*PIIF*). Furthermore, PIIF and FIIF shifts can be mixed together and yield *Mixed Informative Invariant Features* (*MIIF*), as shown in Figure 2. We refer interested readers to Ahuja et al. (2021) for a detailed introduction to the generation process.

**Invariance Learning.** The invariance learning (IL) approach tackles the OOD Generalization problem by predicting invariant features within the data. Considering a classification task, the objective of invariance learning is to find an extractor $\Phi$ such that $\Phi(X^s) = Z_c$ for all $s \in \epsilon_{all}$. The learning objectives for $\Phi$ and $w$ are formulated as:

$$min_{s \in \mathcal{E}_{tr} \subseteq \mathcal{E}_{all}, \Phi, w} R^s(w(\hat{Z}_c); Y) \ s.t. \ \hat{Z}_c \perp s, \ \hat{Z}_c = \Phi(X^s). \tag{1}$$

where $R^s$ is the risk of the function, which is implemented by the cross-entropy loss $\mathcal{L}_{ce} = -\frac{1}{n}\Sigma_{i=1}^n log(\hat{Y}_i^s)Y_i^s$. $\hat{Z}_c \perp s$ is the strong restriction for the model that distinguishes the representation from the interventions

from the environments, only obtaining the information about the invariance $C$. Besides, the basic assumption of the OOD generalization is the environments, which are usually not accessed in real scenarios. So there are essentially two distinct categories of Inverse Learning (IL) methods, depending on whether the environments are explicitly labeled in the training datasets. In this paper, we remove environmental restrictions and focus on environments not covered in the training dataset.

## 3 Over-invariance Issue

### 3.1 Invariant Features Derived from Invariance Principle

The theoretical guarantee to previous IL methods is the invariant principle, which defines what predictor is invariant in different environments. Following Arjovsky et al. (2019), we formally define the invariant principle as follows:

**Definition 3.1 (Invariance Principle)** *We define a data representation function $\Phi$ as eliciting an invariant predictor $w$ across environments $S$ if there is a classifier $w$ simultaneously optimal for all environments. Specifically, this condition can be expressed as follows:*

$$w \in argmin_{\overline{w}}\mathbb{E}(\overline{w} \circ \Phi(X^s)), \ \forall s \in S. \tag{2}$$

The invariant principle gives the rigorous definition of the invariant model. A data representation function $\Phi$ elicits an invariant predictor across environments $S$ if and only if, for all feature $Z$ in the intersection of the supports of $\Phi(X^s)$, we have $\mathbb{E}[Y^s|\Phi(X^s) = Z] = \mathbb{E}[Y^{s'}|\Phi(X^{s'}) = Z]$, for all $s, s' \in S$. For more clarity, we further define the invariant feature derived from the invariant principle.

**Definition 3.2 (Invariant feature)** *Let $I = \{0,1\}^k$ represent a selection function. An invariant feature of label $Y$ under both train distribution $\mathcal{P}^{tr}$ and test distribution $\mathcal{P}^{te}$ is any subset $Z^c = Z \circ I$ of the latent feature $Z \in \mathbb{R}^k$ that satisfies*

$$\mathbb{E}_{p^{tr}}[Y|Z^c] = \mathbb{E}_{p^{tr}}[Y|Z], \ \mathbb{E}_{p^{te}}[Y|Z^c] = \mathbb{E}_{p^{te}}[Y|Z]. \tag{3}$$

An invariant feature, denoted as $Z^c$, consists of features from $X$ that carry predictive power for the target $Y$ across both training and test environments. In other words, $Z^c$ provides as much information about $Y$ as the full feature set $X$, ensuring that the predictive relationship is stable across environments $S$.

### 3.2 Rethinking the Effect of Invariant Features in OOD Generalization

In the above section, we formally define the invariant feature. However, this definition imposes a significant restriction on the feature space, potentially leading to a degradation in out-of-distribution generalization although it alleviates spurious correlations. In this part, we attribute two potential risks of the invariant feature dilemma: 1) limited environmental diversity and 2) over-regularization via the loss function.

**Limited Numbers of Diverse Environments.** A lack of sufficient environmental diversity in real-world scenarios fails to meet the requirements of the invariance principle. According to Equation 2, the hypothesis of Invariant Risk Minimization (IRM) assumes that the environment labels are well-defined and that all environments $\epsilon_{all}$ must be represented in the expectation condition. However, even if the environments in the training set differ, they can still be significantly dissimilar to those in the test set, causing the model to learn shortcuts based on the training data.

**Over-Regularization via Implementation.** In addition to the inherent limitations of the environment collection, there is also a gap between the theory based on the ideal assumption and the implementation in practice. For example, IRMv1 (Arjovsky et al., 2019) employs the $l_2$ norm of the model gradients on the Empirical Risk Minimization (ERM) loss to learn the invariance across training environments. This penalty-based approach is also utilized by various IL methods such as VREx (Krueger et al., 2021), Fishr (Rame

et al., 2022), and IB-IRM (Ahuja et al., 2021). All these methods share the common goal of constraining the rate of feature changes across different environments, preventing overfitting in a specific environment due to the rapid changes in gradients. This strategy aims to force the model to learn the invariant features by stable adjustments in train environments. However, the strong second-order regularization terms in the ERM loss restrict the diversity of invariant features, thereby limiting the model's ability to capture a broad range of relevant features.

The above two reasons highlight the failure cases of the invariance principle, revealing that rough constraints may inadvertently harm valuable invariant features, a phenomenon we refer to as *over-invariance*. Formally, due to the unavailability of test environments $\mathcal{P}^{te}$, such an invariant principle could inadvertently overlook minor invariant characteristics, potentially misinterpreting them as mere hallucinations of spurious features, dubbed as the *over-invariance issue*. In particular, we define the over-invariance issue as follows:

**Definition 3.3 (Over-Invariant feature)** *Let $Z^c$ be the invariant feature of $Y$, if there exist a subset $O^c$ of $Z^c$ that satisfies*

$$\mathbb{E}_{p^{tr}}[Y|O^c] = \mathbb{E}_{p^{tr}}[Y|Z^c], \ \mathbb{E}_{p^{te}}[Y|O^c] \neq \mathbb{E}_{p^{te}}[Y|Z^c], \tag{4}$$

*then $O^c$ is the over-invariant feature.*

An over-invariant feature is the subset of the invariant feature ($O^c \subset Z^c$) that performs similarly in the train environments. While $O^c$ maintains predictive accuracy for $Y$ in the training distribution $\mathcal{P}^{tr}$, it only contains part information of $Y$, may leading to poor OOD performance in $\mathcal{P}^{te}$.

### 3.3 Theoretical Analysis

Since distinguishing between invariant and spurious information in the hidden feature space is challenging in real-world scenarios, we create a synthetic dataset to simulate various distributions, allowing us to further observe the existence of the over-invariance.

**Definition 3.4 (Data Generation)** *Given the data $(\mathbf{x}, y, y_s)$, $y$ is the label and $y_s$ is the environment, $y$ is uniformly sampled from $\{-1, 1\}$ and $y_s = Rad(s) \times y$ where $Rad(s)$ is a random variable taking value $-1$ with with probability $s$ and $1$ with with probability $1 - s$. The data $\mathbf{x} \in \mathbb{R}^d$ is composed of two components: the invariant feature $x_c$ and the spurious feature $x_s$, where $x_c \in \mathbb{R}^{d_c}$, $x_s \in \mathbb{R}^{d_s}$, and $d = d_c + d_s$. Each sample $\mathbf{x}$ is generated as follows:*

$$\mathbf{x} = \{x_c, x_s\} \in \mathbb{R}^d, \ where \begin{cases} x_c \sim N(\mu_c y, \sigma_c^2), \\ x_s \sim N(\mu_s y_s, \sigma_s^2), \end{cases}$$

*Here, $\mu_c \in \mathbb{R}^{d_c}$ and $\mu_s \in \mathbb{R}^{d_s}$ represent the mean of the Gaussian distributions. $\sigma_c \in \mathbb{R}^{d_c \times d_c}$ and $\sigma_s \in \mathbb{R}^{d_s \times d_s}$ denotes the standard deviations that control the variability.*

To analyze the preferences of the invariant learning for different components of invariant features, we quantify the *strength* of the subset of features, by masking the irrelevant data as 0 and measuring their $l_2$ norms of the learned representation. Intuitively, this measures how much information the model extracts from the specified dimensions collectively.

**Definition 3.5 (Strength)** *Given a subset of dimensions $\{m, m+1, \ldots, n\}$, we mask all other dimensions of $\mathbf{x}$ as 0 and pass the masked data $\mathbf{x}_{m:n}$ through the featurizer. Let $\Phi^*$ be the representation function learned by the invariant learning. The strength of the selected feature subset is as follows:*

$$strength(\mathbf{x}_{m:n}) = \|\Phi^*(\mathbf{x}_{m:n})\|_2, \tag{5}$$

In this paper, we set the dimensions of both the invariant and spurious features to $d_c = d_s = 8$. Let $\mathbf{1}_n$ be the all-one vector of length $n$ and **diag** be the diagonal matrix. For the invariant feature $x_c$, we set $\mu_c = 10 * \mathbf{1}_{d_c}$ and $\sigma_c = \mathbf{diag}(5, 5, 3, 3, 1, 1, 0.1, 0.1)$, where different variances represent different important levels of invariance where high variance means more important. For the spurious features $x_s$, we set $\mu_s = 10 * \mathbf{1}_{d_s}$

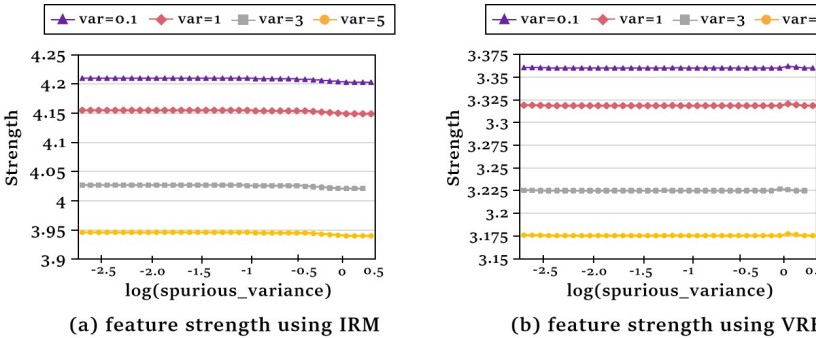

**(a) feature strength using IRM**  **(b) feature strength using VREx**

Figure 3: The strength on different subsets of the invariant feature, lower strength means less preference, verifying the existence of over-invariance issue. The X-axis represents the logarithm of spurious variance $\sigma_s$. The Y-axis shows the strength of the corresponding subset of invariant features $\Phi(x)$ under varying invariant variances $\sigma_c = \{0.1, 1, 3, 5\}$. For each configuration, we run 10 different seeds and report the average results.

and uniformly sample $\sigma_s$ from the range $[10^{-3}, 10^{0.5}]$ simulating noisy environments. We set $s = 0.3$ in the train set and $s = 0.7$ in the test set, representing the OOD environments. We train a two-layer perceptron network featurizer $\Phi : x \to \mathbb{R}^l$ where each layer consists of a linear transformation and a ReLU activation and a one-layer linear classifier predictor $w$. We take the classic invariant learning methods IRMv1 (Arjovsky et al., 2019) and VREx (Krueger et al., 2021) for example.

Figure 3 illustrates the strengths inside the invariant features. We separate the invariant data $x_c$ into 4 parts based on different variances $\{0.1, 1, 3, 5\}$ and calculate their strengths varying with different spurious variances $\sigma_s$ simulating the noisy environments. The results indicate that while all of the features are invariant, their strengths vary. IL methods are selective to invariant features with some invariant features being learned less effectively than others. The lower strength of the subset of the invariant feature suggests that IRMv1 and VREx may struggle with key parts of invariant features, leading to the over-invariance issue. Formally, we give the following informal proposition to further illustrate the over-invariance:

**Remark 3.6 (Over-invariance issue)** *Our synthetic experiment indicates that, with high probability, there exists a subset of the invariant data $x_c$, denoted as the over-invariant data $o_c$, where the strength of the remaining invariant data $(x_c \backslash o_c)$ can vary significantly, potentially approaching zero in extreme cases:*

$$strength(x_c \backslash o_c) > strength(o_c) \to 0.$$

*Thus, **over-invariance** may occur at test time.*

**Over-invariance issue in the real datasets.** We give more visualization examples and discuss them in the interpretation architectures XGNN as shown in Appendix C.2. We have two observations based on the above visualization examples. 1) Invariant learning methods tend to extract the subgraph of the ground truth invariant subgraph, thus verifying the over-invariance issue. 2) When confronted with multiple invariant subgraphs, these learning methods usually focus on only one of them rather than considering all the invariant subgraphs. The above two observations aligned with our synthetics experiments in Figure 3 measured by the strength, thus further verifying the occurrence of the over-invariance issue.

## 4 `DivIL`: Diverse Invariant Learning

Built upon our analysis of the pitfall of the invariant feature and the observation about over-invariance issue, we propose a novel approach Diverse Invariant Learning (`DivIL`) that integrates unsupervised contrastive learning (UCL) and the masking mechanism, which can be applied to various IL methods.

**Enhancing the Environments by Random Data Augmentation** As discussed in Section 3.2, one fundamental assumption of the invariant principle is the requirement for an infinite number of environments,

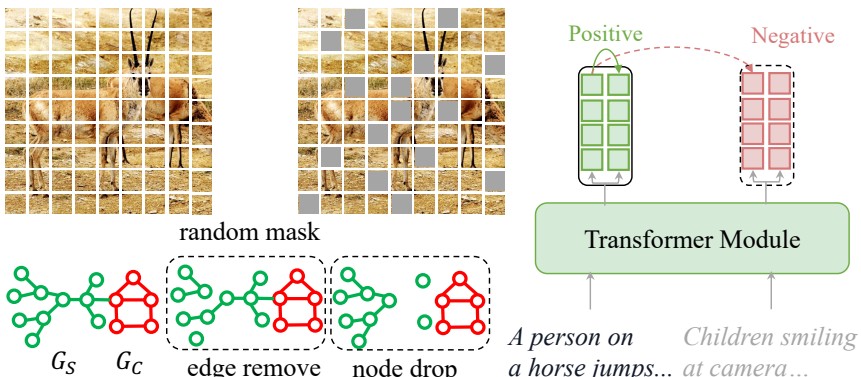

Figure 4: Data augmentation of $\mathcal{L}_{ucl}$ in `DivIL` across multi-modals. Left up: random masking the figure to 0. Left down: edge removing and node dropping for the graph. Right: We feed the same input sequence to the encoder twice by applying different dropout masks to obtain the positive pair.

which is impractical in real-world. To compensate for this limitation, we employ data augmentation to produce a wider range of samples. By introducing variations of data, we disrupt the spurious correlation between the labels and the environments from the train data, fostering the creation of diverse environments. We only use random data augmentation without careful designs which is enough to show the benefits. As illustrated in Figure 4, we use edge dropping, node dropping, and random subgraph extraction for graph following You et al. (2020) and Ding et al. (2022); randomly masking the data $x^{n \times n \times 3}$ to zero with a probability of p for CV, and obtaining $z_i', z_i$ with dropout masks on fully-connected layers as well as attention with a probability of p for NLP (Vaswani et al., 2017; Gao et al., 2021).

**Alleviating the Restriction by Unsupervised Contrastive Learning**  To alleviate the spurious correlations, IL methods usually add strong penalties to the loss (Arjovsky et al., 2019; Krueger et al., 2021; Rame et al., 2022) which tends to suppress subtle yet important details, thus causing the over-invariance issues. Unsupervised learning (UL), particularly unsupervised contrastive learning (UCL) (Oord et al., 2018), provides a powerful mechanism for addressing this by learning the sample-level features, promoting the focus on the minor details of the invariance (Xue et al., 2023a; Jing et al., 2021; Chen et al., 2020a; Qin et al., 2022; Zhang et al., 2022b). Formally, $\mathcal{L}_{ucl}$ is defined as follows:

$$\mathcal{L}_{ucl} = -\sum_{i=1}^{N} \log \frac{\exp(||z_i - z_i'||_2^2)}{\sum_{j \neq i'} \exp(||z_i - z_j||_2^2) + \exp(||z_i - z_i'||_2^2)} \tag{6}$$

Here, $z_i$ represents the feature vector of the original sample, $z_i'$ is the augmented version of the same sample, treated as a positive pair, and $z_j$ is the different sample (negative pair).

**Overall Training Objective of `DivIL`**  By merging our UCL loss with the conventional invariant loss, we aim to balance the diversity of invariant features while preserving the effectiveness of the prior invariant loss in reducing the influence of spurious features. The final objective function of Diverse Invariant Learning (`DivIL`) is as follows:

$$\mathcal{L}_{\texttt{DivIL}} = \mathcal{L}_{pred} + \lambda \mathcal{L}_{il} + \beta \mathcal{L}_{ucl} \tag{7}$$

Here, $\mathcal{L}_{pred}$ is the cross-entropy loss, $\mathcal{L}_{il}$ is any invariant loss from IL methods, and $\mathcal{L}_{ucl}$ is our proposed unsupervised contrastive loss. The hyperparameters $\lambda > 0$ and $\beta > 0$ control the trade-off between invariance and diversity, which can be tuned based on the task. As seen in Figure 5, we observe that after adding the UCL loss, the strength of invariant features increases across various IL penalties, verifying our analysis that incorporating unsupervised contrastive learning as a complement to the invariant loss effectively enhances OOD generalization.

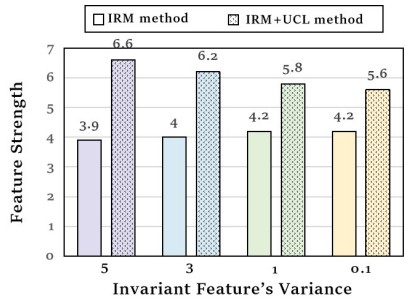 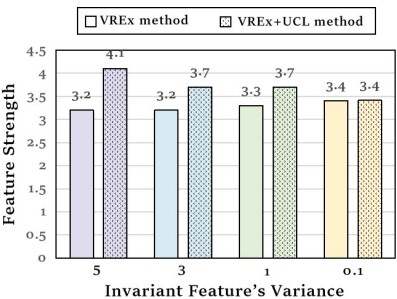

Figure 5: The increased strengths after incorporating UCL with IRMv1 and VREx. The X-axis represents different invariant variances $\sigma_c = \{5, 3, 1, 0.1\}$.The Y-axis shows the strength of the corresponding subset of invariant features $\Phi(x)$ before and after adding the UCL. For each configuration, we run 10 different seeds and report the average results.

---

**Algorithm 1** Overall Training Objective of `DivIL`

---

1: **Input:** Train dataset $\mathcal{D}^{tr} = \{D^s\}_{s \in \epsilon_{tr} \subseteq \epsilon_{all}}$ and test dataset $\mathcal{D}^{te} = \{D^s\}_{s \in \epsilon_{te} \subseteq \epsilon_{all}}$, $\epsilon_{tr} \neq \epsilon_{te}$.
2: **Output:** Trained model $f = w \circ \Phi$
3: **Function:** Representation function $\Phi : X \to H$, Invariant predictor $w : H \to Y$
4: **Hyperparameters:** $\lambda > 0$, $\beta > 0$, mask probability $p$, learning rate $\eta$
5: // `DivIL` OOD Generalization Training
6: **for** each batch $(X_i, Y_i) \in \mathcal{B}$ from $\mathcal{D}^{tr}$ **do**
7:     Calculate cross-entropy loss: $\mathcal{L}_{\text{pred}}$
8:     Calculate invariant loss: $\mathcal{L}_{\text{il}}$
9:     Using data augmentation technique to get the $z$ and $z'$.
10:     masking the front $p$ percent of the entire dimensions of $z$ and $z'$.
11:     Calculate unsupervised contrastive loss: $\mathcal{L}_{\text{ucl}}$ in equation 6.
12:     Calculate total loss: $\mathcal{L}_{\text{DivIL}} = \mathcal{L}_{\text{pred}} + \lambda \mathcal{L}_{\text{il}} + \beta \mathcal{L}_{\text{ucl}}$
13:     Compute gradients: $\nabla_\Phi = \frac{\partial \mathcal{L}_{\text{DivIL}}}{\partial \Phi}, \nabla_w = \frac{\partial \mathcal{L}_{\text{DivIL}}}{\partial w}$
14:     Update model parameters:
15:         $\Phi \leftarrow \Phi - \eta \nabla_\Phi$
16:         $w \leftarrow w - \eta \nabla_w$
17: **end for**

---

**Remark 4.1 (Effectiveness of `DivIL`)** *Our synthetic experiment indicates that for the original over-invariant data $o_c$ illustrated in Remark 3.1, the strength of $o_c$ learned by* **DivIL** *in equation 7 is bigger than the original invariant learning:*

$$strength_{DivIL}(o_s) > strength_{IRM/VREx}(o_s),.$$

*Thus,* **DivIL** *effectively mitigates the* ***over-invariance*** *issue at test time.*

**Enhancing Diversity of the Invariance via Random Masking**   Contrastive learning methods, by repelling negative samples, can alleviate the over-invariance problem to some extent. However, when faced with strong data augmentation or deep-layer implicit regularization, the model performance can also remain suboptimal (Jing et al., 2021). To further enhance feature diversity, we trained a non-linear projector to scatter the representation space spectrum. Additionally, we introduced a random masking mechanism to the features to overcome over-invariance. We set the first p dimensions of the contrastive learning feature dimension to 0, that is $z_{1:p} = 0$.

In conclusion, the detailed training procedure of `DivIL` is shown in Algorithm 1.

Table 1: Performance on real-world graph dataset, demonstrating better OOD generalization ability of `DivIL`. The blue, gray, and Underline highlight the first, second, and third best results, respectively. All results are reported with mean $\pm$ std across seeds $\{0, 1, 2, 3, 4\}$.

| | EC50-Assay | EC50-Scaffold | EC50-size | Ki-Assay | Ki-Scaffold | Ki-size | CMNIST-sp | Graph-SST2 |
|---|---|---|---|---|---|---|---|---|
| ERM | $70.30 \pm 2.15$ | $63.45 \pm 1.43$ | $61.47 \pm 1.99$ | $70.43 \pm 2.19$ | $72.43 \pm 1.38$ | $71.43 \pm 3.60$ | $25.67 \pm 9.70$ | $82.75 \pm 0.20$ |
| IRM | $71.00 \pm 4.47$ | $60.42 \pm 0.69$ | $60.30 \pm 1.18$ | $70.39 \pm 1.44$ | $69.38 \pm 2.81$ | $70.80 \pm 2.63$ | $19.19 \pm 2.83$ | $82.31 \pm 1.22$ |
| VREX | $71.91 \pm 6.68$ | $62.07 \pm 1.30$ | $61.03 \pm 1.27$ | $68.74 \pm 4.13$ | $70.51 \pm 3.13$ | $70.34 \pm 4.26$ | $14.91 \pm 1.85$ | $82.40 \pm 0.63$ |
| EIIL | $70.39 \pm 3.11$ | $61.20 \pm 1.68$ | $60.31 \pm 1.64$ | $69.20 \pm 2.29$ | $69.99 \pm 1.58$ | $72.78 \pm 3.08$ | $22.37 \pm 7.35$ | $82.31 \pm 1.50$ |
| IB-IRM | $67.04 \pm 2.66$ | $61.04 \pm 1.13$ | $62.20 \pm 0.64$ | $71.94 \pm 2.42$ | $74.16 \pm 1.29$ | $71.15 \pm 4.44$ | $37.44 \pm 7.36$ | $81.95 \pm 0.74$ |
| CNC | $74.96 \pm 2.48$ | $63.59 \pm 0.87$ | $60.44 \pm 2.15$ | $74.08 \pm 3.67$ | $67.54 \pm 1.26$ | $68.15 \pm 5.24$ | $19.41 \pm 3.15$ | $80.72 \pm 1.15$ |
| CNCP | $73.74 \pm 2.62$ | $62.05 \pm 1.22$ | $60.53 \pm 2.14$ | $74.13 \pm 2.46$ | $67.70 \pm 2.85$ | $67.54 \pm 3.37$ | $24.99 \pm 4.70$ | $80.76 \pm 0.64$ |
| CIGA | $76.63 \pm 1.16$ | $66.25 \pm 1.49$ | $63.66 \pm 1.15$ | $71.55 \pm 1.84$ | $71.54 \pm 2.48$ | $74.52 \pm 3.09$ | $41.35 \pm 5.56$ | $82.89 \pm 0.97$ |
| DivIL | $77.00 \pm 1.52$ | $67.41 \pm 0.55$ | $64.33 \pm 0.83$ | $72.64 \pm 2.78$ | $73.38 \pm 0.91$ | $75.99 \pm 2.46$ | $46.29 \pm 11.2$ | $83.30 \pm 0.91$ |

## 5 Experiments

We evaluate `DivIL` and compare with IL methods on a range of tasks requiring OOD generalization. `DivIL` provides generalization benefits and outperforms IL methods on a wide range of tasks, including: 1) graph datasets such as the synthetic Spurious-Motif and drug discovery, 2) Colored MNIST (CMNIST) dataset, and 3) natural language datasets.

### 5.1 Experiments on Graph

**Datasets** We employed one synthetic dataset along with eight realistic datasets, including the Spurious-Motif datasets introduced in Wu et al. (2022). These datasets consist of three graph classes, each containing a designated subgraph as the ground-truth explanation. Additionally, there are spurious correlations between the remaining graph components and the labels in the training data, challenging the model's ability to differentiate between true and spurious features. The degree of these correlations is controlled by the parameter $b$, with values of $0.33, 0.6$, and $0.9$. Furthermore, to examine our method in real-world scenarios characterized by more complex relationships and distribution shifts, we incorporated the DrugOOD dataset (Ji et al., 2022) from AI-aided Drug Discovery, which includes Assay, Scaffold, and Size splits from the EC50 category (denoted as EC50-*) and the Ki category (denoted as Ki-*). Additionally, we included tests on the CMNIST-sp dataset, which consists of superpixel graphs derived from the ColoredMNIST dataset using the algorithm from Knyazev et al. (2019), featuring distribution shifts in node attributes and graph size. We also tested on the sentiment analysis dataset Graph-SST2 (Yuan et al., 2020), which is formed by converting each text sequence from SST2 into a graph representation.

**Baselines** We compared `DivIL` with causality-inspired invariant graph learning methods, such as IRM Arjovsky et al. (2019), v-Rex Krueger et al. (2021), and IB-IRM Ahuja et al. (2021). Additionally, we evaluated `DivIL` against methods like EIIL Creager et al. (2021b), CNC, CNCP Zhang et al. (2022a), and CIGA Chen et al. (2022), all of which do not require environment labels. Notably, CNC, CNCP, and CIGA employ contrastive sampling strategies to address the OOD problem. We implemented $\mathcal{L}_{il}$ following the SOTA method CIGA, and for $\mathcal{L}_{ucl}$, we selected the best-performing DA techniques, such as edge removal, node dropping, and subgraph extraction, based on You et al. (2020). We report classification accuracy for the Spurious-Motif, CMNIST-sp, and Graph-SST2 datasets, and ROC-AUC for the DrugOOD datasets. The evaluation is conducted five times with different random seeds ($\{1, 2, 3, 4, 5\}$), selecting models based on validation performance. We utilized the GCN backbone Kipf & Welling (2017) with sum pooling to enhance across all experiments.

**`DivIL` outperforms previous IL methods.** As demonstrated in Table 1, `DivIL` shows better generalization ability than all baseline models on real-world datasets. Specifically, in the MNIST-sp dataset, `DivIL` surpasses CIGA by 5%. Furthermore, in the ki-scaffold and ki-assay datasets, CIGA performs worse than ERM, while `DivIL` by implementing the $\mathcal{L}_{il}$ on CIGA achieves higher performance. The results not only highlight the competitive edge of `DivIL` over established baselines but also emphasize its generalization across

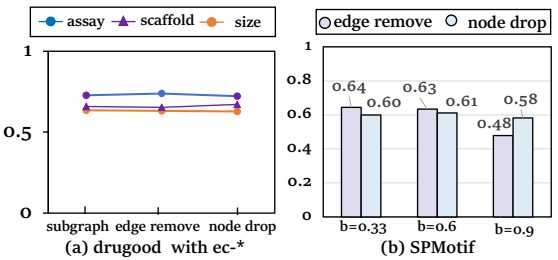

Figure 6: Ablation study of different graph data augmentations in `DivIL`. (a) compares the performance of subgraph extraction, edge removing, and node dropping on the ec-* category from DrugOOD datasets. (b) illustrates the performance of edge removing and node dropping on SPMotif with different OOD shift biases.

Table 2: Performance on Spurious-Motif dataset with different GNN backbones. `DivIL` can enhance the performance on various architectures. The blue, gray, and Underline highlight the first, second, and third best results, respectively. All results are reported with mean $\pm$ std across seeds $\{0, 1, 2, 3, 4\}$.

|  | SPMotif-0.33 | SPMotif-0.60 | SPMotif-0.90 |
|---|---|---|---|
| GNN | | | |
| GIN | $47.8 \pm 8.03$ | $49.21 \pm 4.2$ | $44.11 \pm 5.5$ |
| $+\mathcal{L}_{il}$ | $50.67 \pm 3.83$ | $50.39 \pm 2.19$ | $42.35 \pm 6.36$ |
| +`DivIL` | $51.48 \pm 5.43$ | $51.77 \pm 4.89$ | $45.84 \pm 3.82$ |
| GCN | $58.51 \pm 2.84$ | $50.51 \pm 4.75$ | $44.67 \pm 3.5$ |
| $+\mathcal{L}_{il}$ | $67.53 \pm 1.35$ | $59.96 \pm 8.59$ | $47.66 \pm 6.08$ |
| +`DivIL` | $67.81 \pm 3.33$ | $62.79 \pm 3.86$ | $50.93 \pm 9.05$ |
| XGNN | | | |
| $+\mathcal{L}_{il}$(GIN) | $57.58 \pm 3.73$ | $58.11 \pm 4.29$ | $52.14 \pm 3.27$ |
| +`DivIL` (GIN) | $63.86 \pm 2.19$ | $60.31 \pm 2.04$ | $52.54 \pm 6.57$ |
| $+\mathcal{L}_{il}$(GCN) | $63.37 \pm 4.27$ | $65.45 \pm 4.91$ | $59.64 \pm 4.64$ |
| +`DivIL` (GCN) | $63.90 \pm 3.7$ | $70.03 \pm 3.66$ | $66.85 \pm 6.61$ |

varying datasets. Such nuanced performance differentials underscore our capabilities of `DivIL` in navigating complex real-world datasets, positioning the over-invariance issues as a crucial problem.

**`DivIL` shows effectiveness on various backbones.** Additionally, we incorporate XGNN, an interpretable GNN to extract the invariant subgraph $G_c$ commonly used in graph OOD models (Wu et al., 2022; Li et al., 2022; Chen et al., 2022). Specifically, a XGNN $w_x \circ \Phi_x$ is with an extractor $\Phi_x : \mathcal{G} \to \mathcal{G}$ that identifies an invariant subgraph $G_c$ to help predict their labels $y_x = w_x(G_c)$ with a downstream classifier $w : \mathcal{G} \to \mathcal{Y}$. Table 2 shows that `DivIL` significantly outperforms Vanilla GNN and IL (we implement the IL methods with one of the SOTA graph IL methods CIGA (Chen et al., 2022)) on Spurious-Motif under various backbones like GCN, GIN, and XGNN settings. Moreover, as the spurious bias increases, the performance of `DivIL` remains more stable, while the baselines and IL models tend to fail, like in the SPMotif-0.60 dataset `DivIL` improves performance from 65.45% to 70.03% and in the SPMotif-0.90 dataset from 59.64% to 66.85%.

**Sensitivity on different graph data augmentations.** Figure 6(a) illustrates that different random augmentation methods, such as subgraph extraction, edge removing, and node dropping (You et al., 2020), yield similar performance in addressing graph out-of-distribution (OOD) challenges, echoing observations found in recent studies (Guo et al., 2023). Additionally, in Figure 6(b), the comparison between edge remoing and node dropping methods in SPMotif under varying shift biases reveals a slight advantage of edge removing over node dropping at $b = 0.33$ and $b = 0.6$. However, at $b = 0.9$, node dropping surpasses edge removal, although the difference remains modest. This observation supports our insight that the data augmentation strategies employed may not significantly influence the graph OOD problems.

## 5.2 Experiments on CMNIST

We evaluate `DivIL` on the synthetic datasets ColoredMNIST following Arjovsky et al. (2019). We compare `DivIL` with ERM, and various IL methods, including causal methods that focus on learning invariance (IRM, VREx) and gradient matching techniques (Fishr). As previously done in Fishr, we maintain all IL method implementations identical to the IRM implementation, notably the same MLP and hyperparameters, and just add the `DivIL` penalty to the loss. We use two-stage scheduling selected in IRM for the regularization strength $\lambda$, which is low until epoch 190 and then jumps to a large value. Due to the varying degrees of over-invariance introduced by different IL methods, we performed a simple search over $\beta$ values of $\{0.01, 0.05, 0.1, 0.2\}$, and mask probabilities $p$ of $\{0.3, 0.5, 0.7\}$.

Table 3 reports the accuracy averaged over 5 runs with standard deviation. Adding `DivIL` can achieve the trade-off between train and test accuracies, notably in test set. It reaches 69.25% in the colored test set

and 70.43% when digits are grayscale. In addition, `DivIL` improves the performance in all IL methods, verifying our understanding of the issue of over-invariance. Figure 7(a) displays the results of `DivIL` using different invariant losses across various mask percentages $p$, demonstrating the robustness of `DivIL` to the hyperparameter $p$ with minimal variance in accuracy across different $p$ values. Figure 7(b) illustrates that increasing the weight $\beta$ of the $\mathcal{L}_{ucl}$ term leads to improved performance in IRMv1, VREx, and Fishr. This supports our insight that over-invariance issues exist in current incremental learning (IL) methods. By introducing diversity penalties $\mathcal{L}_{ucl}$, we can mitigate this issue and enhance out-of-distribution (OOD) performance.

Table 3: Performance on CMNIST dataset. All results are reported with mean $\pm$ std over 5 runs. DivIL improves the accuracy on the test and the gray set, demonstrating its effectiveness in alleviating over-invariance issues in the visual domain.

|  | train set | test set | gray set |
|---|---|---|---|
| ERM | $86.47 \pm 0.16$ | $14.18 \pm 0.68$ | $70.74 \pm 0.77$ |
| IRM | $71.47 \pm 1.18$ | $65.30 \pm 1.09$ | $66.66 \pm 2.33$ |
| + `DivIL` | $70.93 \pm 0.29$ | $66.40 \pm 1.39$ | $66.97 \pm 1.85$ |
| VREx | $72.14 \pm 1.49$ | $67.05 \pm 0.84$ | $68.96 \pm 2.03$ |
| + `DivIL` | $72.67 \pm 0.93$ | $67.50 \pm 1.45$ | $69.30 \pm 1.91$ |
| Fishr | $71.34 \pm 1.27$ | $69.18 \pm 0.80$ | $70.35 \pm 1.14$ |
| + `DivIL` | $71.27 \pm 1.36$ | $69.25 \pm 0.81$ | $70.43 \pm 1.02$ |

### 5.3 Experiments on Natural Language Inference

Table 4: Performance on SNLI(in-domain), MNLI matched and mismatched (out-domain) dataset. DivIL effectively enhances out-of-distribution generalization while preserving in-domain test performance. Additionally, it provides an ablation study demonstrating the necessity of both $\mathcal{L}_{ucl}$ and $\mathcal{L}_{il}$ components.

|  | SNLI | MNLI | |
|---|---|---|---|
|  |  | matched | mismatched |
| ERM | 77.7 | 54.4 | 54.7 |
| IRM | 77.7 | 55.0 | 55.5 |
| `DivIL` | 79.3 | 55.5 | 59.2 |
| `DivIL`- $\mathcal{L}_{ucl}$ | 77.6 | 54.5 | 56.6 |
| `DivIL`- $\mathcal{L}_{il}$ | 78.8 | 54.5 | 57.1 |

Inspired by Qin et al. (2024), we also demonstrated the effectiveness of our method in NLP through a Natural Language Inference (NLI) (Dagan et al., 2013) task, which assesses the logical relationship between two sentences: entailment, contradiction, or neutrality. Our model was trained on a subset of the SNLI (Bowman et al., 2015) training set and evaluated on selected cases from the SNLI validation set, as well as the match and mismatch subsets of the MNLI (Williams et al., 2017) validation set. While SNLI represents an in-distribution (ID) scenario, MNLI helps assess the generalization to out-of-distribution (OOD) data. The results show that our method performs well in both IID and OOD scenarios, validating its effectiveness.

In our experiment, we employed a pretrained GPT-2 model with a randomly initialized classification head. We set the maximum token length to 64 and trained the model for 5 epochs using the AdamW optimizer. The learning rate was configured at 2e-5, with a weight decay of 0.01 and a linear learning rate scheduler. We used a training batch size of 32. To optimize our model, we implemented supervised contrastive loss as $\mathcal{L}_{il}$ following Zhang et al. (2022a) and explored various combinations of weights for $\lambda$ and $\beta$, choosing values from the set $\{0, 0.1, 0.3, 0.5, 0.7, 1.0\}$. Additionally, we fixed the projection mask probability at 0.7 and reported the results for the best-performing configuration.

Table 4 shows the results of `DivIL` on the NLI task, where `DivIL` outperforms both IRM and ERM approaches on real-world natural language datasets. Furthermore, our ablation study reveals that removing either $\mathcal{L}_{il}$

or $\mathcal{L}_{ucl}$ leads to a decrease in OOD performance. However, the performance remains better than that of IRM or ERM, indicating that `DivIL` achieves a trade-off between strong regularization and feature diversity. Figure 7(c) illustrates the performance of `DivIL` with varying weights, denoted as $\beta$, for the loss function $\mathcal{L}_{ucl}$ across the SNLI, MNLI-match, and MNLI-mismatch datasets. Unlike the findings in CMNIST, increasing $\beta$ does not necessarily improve out-of-distribution (OOD) performance. It's essential to choose an appropriate weight, possibly due to the unique structure of natural language.

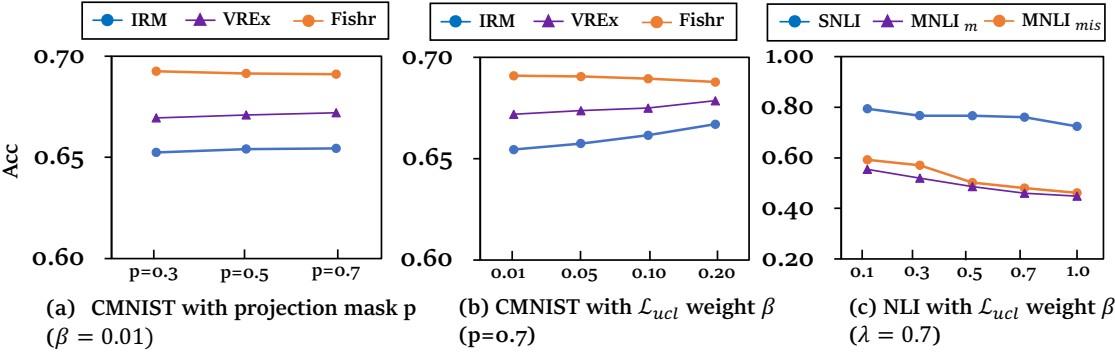

(a) CMNIST with projection mask p
($\beta = 0.01$)

(b) CMNIST with $\mathcal{L}_{ucl}$ weight $\beta$
(p=0.7)

(c) NLI with $\mathcal{L}_{ucl}$ weight $\beta$
($\lambda = 0.7$)

Figure 7: Ablation study of `DivIL` on CMNIST and NLI datasets. (a) illustrates the performance of `DivIL` on various mask percent $p$ across different implementation of $\mathcal{L}_{il}$. (b) and (c) illustrates the performance of `DivIL` on different weight of UCL $\beta$ in CMNIST and NLI, respectively.

## 6 Related work

**Out-of-Distribution (OOD) Generalization.** Existing strategies to tackle OOD generalization can be broadly classified into three approaches (Yang et al., 2024). Representation learning focuses on developing robust feature representations that generalize across various distributions, including unsupervised domain generalization (Mahajan et al., 2021; Zhang et al., 2022b; Chen et al., 2020b) and disentangled representations (Bengio et al., 2013; Higgins et al., 2017; Kim & Mnih, 2018; Yang et al., 2021). Model-based approaches, such as Invariant Learning (Rosenfeld et al., 2020; Ganin & Lempitsky, 2015; Li et al., 2018; Creager et al., 2021a) and causal learning (Peters et al., 2016a; Pfister et al., 2019), aim to capture invariant relationships across environments to enhance robustness against distributional shifts. Finally, optimization-based techniques seek to ensure strong worst-case performance under potential distributional changes (Delage & Ye, 2010; Namkoong & Duchi, 2016; Duchi & Namkoong, 2021; Duchi et al., 2023; Zhou et al., 2022), safeguarding models from uncertainties in the data.

**Discussion on Different Invariant Losses** Many invariant learning methods focus on learning stable features across environments by incorporating penalties into the loss function to ensure the consistent change rate (Arjovsky et al., 2019; Krueger et al., 2021; Rame et al., 2022; Zhang et al., 2022a). One series of methods relies on explicit environment labels. For example, IRMv1 (Arjovsky et al., 2019) implements the theory of the invariance principle in practice by assuming the classifier as the constant and employing a gradient-based penalty that requires the sum of the gradients of the model to remain small. The loss function $\mathcal{L}_{irmv1}$ is defined as follows: $\mathcal{L}_{irmv1} = \Sigma_{s \in \epsilon_{tr}} \| \nabla_{w|w=1.0} \mathcal{L}_{pred}^s(f) \|_2$. VREx (Krueger et al., 2021) takes the variance of the loss across different environments defined as $\mathcal{L}_{vrex} = Var(\mathcal{L}_{pred}^1(f), \mathcal{L}_{pred}^2(f), \cdots, \mathcal{L}_{pred}^k(f))$, where k is the environment numbers in the training dataset.

There are also plentiful studies in invariant learning without environment labels. Creager et al. (2021b) proposed a minmax formulation to infer the environment labels. Liu et al. (2021b) proposed a self-boosting framework based on the estimated invariant and variant features. Liu et al. (2021a); Zhang et al. (2022a) proposed to infer labels based the predictions of an ERM trained model. Other methods adopt the loss of supervised contrastive learning as $\mathcal{L}_{il}$, like CNC Zhang et al. (2022a) and CIGA Chen et al. (2022), using different heuristic strategies to choose the positive and negative samples. For example, the invariant penalty

of CIGA encodes the subgraph $G_c$ and take the learned feature $z_k$ from the same label, treated as the positive pair, and $z_j$ is the from different labels (negative pair).

**Contrastive Learning.** SimCLR(Chen et al., 2020a) and MoCo(He et al., 2020) demonstrate how contrastive objectives can improve feature robustness and help models generalize better to unseen environments. Additionally, (Wen & Li, 2021) and (Ji et al., 2023) demonstrate that contrastive learning can effectively extract semantically meaningful features from data. Furthermore, (Xue et al., 2023b) conducts systematic experiments on contrastive learning, revealing the effectiveness of combining supervised and unsupervised contrastive learning for feature learning. By clustering similar features and pushing apart dissimilar ones, contrastive learning prevents feature collapse, even under strong regularization (Chen et al., 2022; Zhang et al., 2022a).

## 7 Conclusion

We shed light on the limitations of invariant constraints in addressing out-of-distribution generalization. While these constraints can mitigate spurious correlations, our research revealed the risk of *over-invariance*, potentially leading to the loss of crucial details in invariant features and a subsequent decline in generalization performance. To tackle these challenges, we introduced Diverse Invariant Learning (`DivIL`), leveraging contrastive learning and random feature masking to introduce uncertainty and diversity. Our comprehensive experiments spanning various modalities and models, underscored the efficacy of our proposed method in enhancing model performance.

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

## A  Datasets

### A.1  CMNIST

Colored MNIST is a binary digit classification dataset introduced in IRM (Arjovsky et al. (2019)). Compared to the traditional MNIST, it has 2 main differences. First, 0-4 and 5-9 digits are each collapsed into a single class, with a 25% chance of label flipping. Second, digits are either colored red or green, with a strong correlation between label and color in training. However, this correlation is reversed at test time. Specifically, in training, the model has access to two domains E = {90%, 80%}: in the first domain, green digits have a 90% chance of being in 5-9; in the second, this chance goes down to 80%. In test, green digits have a 10% chance of being in 5-9. Due to this modification in correlation, a model should ideally ignore the color information and only rely on the digits' shape: this would obtain a 75% test accuracy.

### A.2  NLI

The natural language inference (NLI) task involves determining the logical relationship between pairs of sentences, typically categorized as entailment, contradiction, or neutral. In this task, a model is presented with a premise sentence and a hypothesis sentence, and it must infer how the hypothesis relates to the premise as seen in Table 6 and Table 7. NLI is crucial in natural language understanding as it tests a model's ability to comprehend and reason about language, making it a fundamental benchmark for evaluating the performance of language models and their ability to capture semantic relationships and contextual information within the text.

We provide more details about the motivation and construction method of the datasets used in our experiments. Statistics of the datasets are presented in Table 5. We use about 8,000 examples in the train set from the SNLI Bowman et al. (2015), from the Image Captions from the Flickr30k Corpus domains. We selected 1,000 examples from the validation-matched set of the MNLI dataset (Williams et al., 2017), sourced from the Fiction, Government, Slate, Telephone, and Travel domains. Additionally, we chose another 1,000 examples from the validation-matched set of the MNLI dataset, taken from the 9/11, Face-to-Face, Letters, OUP, and Verbatim domains, to form our out-of-domain (OOD) test set. Examples of SNLI and MNLI are shown in Table 6 and Table 7.

Table 5: Statistics of our constructed OOD NLI Dataset.

| Split | Genre | Examples | Partition | Data Domain | Metrics |
|---|---|---|---|---|---|
| Train set | SNLI | 7992 | train | IMAGE CAPTIONS FROM THE FLICKR30K CORPUS | ACC |
| Test set | SNLI | 991 | validation | IMAGE CAPTIONS FROM THE FLICKR30K CORPUS | ACC |
| | MNLI | 1000 | validation-matched | FICTION, GOVERNMENT, SLATE, TELEPHONE, TRAVEL | ACC |
| | | 1000 | validation-mismatched | 9/11, FACE-TO-FACE, LETTERS, OUP, VERBATIM | ACC |

### A.3  Graph

We provide more details about the motivation and construction method of the datasets used in our experiments. Statistics of the datasets are presented in Table 8.

**Spurious-Motif**  We construct 3-class synthetic datasets based on BAMotif following Wu et al. (2022), where the model needs to tell which one of three motifs (House, Cycle, Crane) the graph contains. For each dataset, we generate 3,000 graphs for each class in the training set, and 1,000 graphs for each class in the validation set and testing set, respectively. We introduce the bias based on FIIF, where the motif and one of the three base graphs (Tree, Ladder, Wheel) are artificially (spuriously) correlated with a probability of various biases, and equally correlated with the other two. Specifically, given a predefined bias b, the

Table 6: NLI dataset samples from SNLI (IID).

| Premise | Hypothesis | Label |
|---|---|---|
| Two men holding their mouths open. | Two men with mouths agape. | ENTAILMENT |
| Trying very hard not to blend any of the yellow paint into the white. | Someone is painting a house. | NEUTRAL |
| A man on a small 4 wheeled vehicle is flying through the air. | The man is on a bike. | CONTRADICTION |
| Two power walkers walking beside one another in a race. | Two people in a park walking | NEUTRAL |
| Women standing at a podium with a crowd and building in the background. | woman stands at podium | ENTAILMENT |

Table 7: MNLI dataset samples from the validation-matched (above) and validation-mismatched (below) subsets (OOD).

| Premise | Hypothesis | Label |
|---|---|---|
| pretty good newspaper uh-huh | I think this is a decent newspaper. | ENTAILMENT |
| Massive tidal waves swept over Crete, and other parts of the Mediterranean, smashing buildings and drowning many thousands of people. | The waves came with no warning to the inhabitants. | NEUTRAL |
| For such a governmentwide review, an entrance conference is generally held with applicable central agencies, such as the Office of Management and Budget (OMB) or the Office of Personnel Management. | An entrance conference is held with specialized agencies. | CONTRADICTION |
| As Figure 6.6 shows, the safety stock needed to achieve a given customer service level is proportional to the standard deviation of the demand forecast. | Figure 6.6 shows the safety stock needed to achieve a given customer service level. | ENTAILMENT |
| Some of Bin Ladin's close comrades were more peers than subordinates. | There were three people who could be considered peers of Bin Ladin. | NEUTRAL |
| Nothing except knowing that you are helping to protect the Earth's precious natural resources. | Everything, except knowing that you are helping to protect Earth's natural resources. | CONTRADICTION |

probability of a specific motif (e.g., House) and a specific base graph (Tree) will co-occur is $b$ while for the others is $(1-b)/2$ (e.g., House-Ladder, House-Wheel). We use random node features in order to study the influences of structure level shifts.

**CMNIST-sp**  To study the effects of PIIF shifts, we select the ColoredMNIST dataset created in Arjovsky et al. (2019). We convert the ColoredMNIST into graphs using the superpixel algorithm introduced by Knyazev et al. (2019) .

**Graph-SST2**  Inspired by the data splits generation for studying distribution shifts on graph sizes, we split the data curated from sentiment graph data [84], that converts sentiment sentence classification datasets Graph-SST2 (Yuan et al., 2020) into graphs, where node features are generated using BERT (Devlin et al., 2019) and the edges are parsed by Gardner et al. (2018). Our splits are created according to the averaged degrees of each graph. Specifically, we assign the graphs as follows: Those that have smaller or equal to 50-th percentile while smaller than 80-th percentile are assigned to the validation set, and the left are assigned to test set.

**DrugOOD datasets** To evaluate the OOD performance in realistic scenarios with realistic distribution shifts, we also include three datasets from DrugOOD benchmark (Ji et al., 2022). DrugOOD is a systematic OOD benchmark for AI-aided drug discovery, focusing on the task of drug target binding affinity prediction for both macromolecule (protein target) and smallmolecule (drug compound). The molecule data and the notations are curated from realistic ChEMBL database (Mendez et al., 2019). Complicated distribution shifts can happen on different assays, scaffolds and molecule sizes. In particular, we select DrugOOD-lbap-core-ec50-assay, DrugOOD-lbap-core-ec50-scaffold, DrugOOD-lbap-core-ec50-size, DrugOOD-lbap-core-ki-assay, DrugOOD-lbap-core-ki-scaffold, and DrugOOD-lbap-core-ki-size, from the task of Ligand Based Affinity Prediction which uses ic50 measurement type and contains core level annotation noises. We directly use the data files provided by Ji et al. (2022).

Table 8: Graph dataset details. The number of nodes and edges are respectively taking average among all graphs.

| Dataset | Training | Validation | Testing | Classes | Nodes | Edges | Metrics |
|---|---|---|---|---|---|---|---|
| SPMOTIF | 9,000 | 3,000 | 3,000 | 3 | 44.96 | 65.67 | ACC |
| CMNIST-SP | 40,000 | 5,000 | 15,000 | 2 | 56.90 | 373.85 | ACC |
| Graph-SST2 | 24,881 | 7,004 | 12,893 | 2 | 10.20 | 18.40 | ACC |
| EC50-Assay | 4,978 | 2,761 | 2,725 | 2 | 40.89 | 87.18 | ROC-AUC |
| EC50-Scaffold | 2,743 | 2,723 | 2,762 | 2 | 35.54 | 75.56 | ROC-AUC |
| EC50-Size | 5,189 | 2,495 | 2,505 | 2 | 35.12 | 75.30 | ROC-AUC |
| Ki-Assay | 8,490 | 4,741 | 4,720 | 2 | 32.66 | 71.38 | ROC-AUC |
| Ki-Scaffold | 5,389 | 4,805 | 4,463 | 2 | 29.96 | 65.11 | ROC-AUC |
| Ki-Size | 8,605 | 4,486 | 4,558 | 2 | 30.35 | 66.49 | ROC-AUC |

## B Implement Details

During the experiments, we do not tune the hyperparameters exhaustively while following the common recipes for optimizing the models. Details are as follows. We will publish our code when the paper is accepted.

### B.1 CMNIST Implements

In the experimental setup in Section 5.2, the network is a 3 layers MLP with ReLu activation, optimized with Adam (Kingma & Ba (2015)). IRM selected the following hyperparameters by random search over 50 trials: hidden dimension of 390, l2 regularizer weight of 0.00110794568, learning rate of 0.0004898536566546834, penalty anneal iters (or warmup iter) of 190, penalty weight ($\lambda$) of 91257.18613115903, 501 epochs and batch size 25,000 (half of the dataset size). For the implementation of the invariant losses(IRM, VREx and Fishr), we strictly keep the same hyperparameters values in our implementation and the code is almost unchanged from `https://github.com/alexrame/fishr`. To account for the varying degrees of over-invariance introduced by different IL methods, we performed a straightforward search over $\beta$ values of {0.01, 0.05, 0.1, 0.2} and projection mask probabilities of {0.3, 0.5, 0.7}, while keeping the random augmentation mask probability fixed at 0.2.

### B.2 NLI Implements

We employed a pretrained GPT-2 model with a randomly initialized classification head. We set the maximum token length to 64 and trained the model for 5 epochs using the AdamW optimizer. The learning rate was configured at 2e-5, with a weight decay of 0.01 and a linear learning rate scheduler. We used a training batch size of 32. To optimize our model, we explored various combinations of the invariant loss and unsupervised loss weights for $\lambda$ and $\beta$, choosing the best from the {0, 0.1, 0.3, 0.5, 0.7, 1.0} according to the validation set. Additionally, we fixed the projection mask probability at 0.7 and reported the results for the best-performing configuration.

### B.3  Graph Implements

For a fair comparison, `DivIL` uses the same GNN architecture for GNN encoders as the baseline methods. We use the GCN backbone and the sum pooling in Table 1. By default, we fix the temperature to be 1 in the unsupervised contrastive loss, and merely search the penalty weight of the contrastive loss from $\{0.1, 0.2, 0.5, 1, 2\}$ according to the validation performances. We select the best of the random mask percentage $p$ from the $\{0.2, 0.3, 0.5, 0.7\}$ according to the validation performances. For the implementation of graph data augmentation, we use the tool from You et al. (2020). We select the best percentage $p_2$ of node dropping, edge removing, and subgraph extraction from the $\{0.05, 0.1, 0.15, 0.2\}$ according to the validation performances to create the positive pair and keep $p_1 = 0$ representing the sample itself. For the implementation of our baselines, we take the code almost unchanged from `https://github.com/LFhase/CIGA`.

## C  More Ablation Study

### C.1  Abltions on Random Masking

We introduced a random masking mechanism to overcome the issue of over-invariance, which differs from the traditional dropout method. Unlike dropout, which randomly drops out dimensions independently, we observed that features tend to diversify in the earlier dimensions and become more invariant in the later dimensions. Therefore, we promote the diversity of characteristics by setting the first p dimensions of the contrastive learning feature space to 0. This approach not only alleviates the over-invariance problem but also enhances flexibility in specific dimensions of the feature space.

To validate the effectiveness of this mechanism, we conducted ablation experiments on the CMNIST dataset. The experimental results showed that the model using the random masking method outperforms the model using traditional dropout in terms of out-of-distribution generalization. As shown in Table 9, the random masking mechanism improves the robustness of the model, particularly in the face of complex data distributions, effectively mitigating over-invariance and improving the overall performance of the model.

Table 9: Comparison of Random Masking and Dropout on the CMNIST Dataset.

| Method | Train | Test |
|---|---|---|
| IRM | $71.47 \pm 1.18$ | $65.30 \pm 1.09$ |
| + `DivIL` (dropout) | $70.65 \pm 0.54$ | $66.19 \pm 1.78$ |
| + `DivIL` (random masking) | $70.93 \pm 0.29$ | $66.40 \pm 1.39$ |
| VERx | $72.14 \pm 1.49$ | $67.05 \pm 0.84$ |
| + `DivIL` (dropout) | $72.54 \pm 1.72$ | $66.73 \pm 1.34$ |
| + `DivIL` (random masking) | $72.67 \pm 0.93$ | $67.50 \pm 1.45$ |
| Fishr | $71.34 \pm 1.27$ | $69.18 \pm 0.80$ |
| + `DivIL` (dropout) | $71.36 \pm 1.41$ | $69.20 \pm 0.77$ |
| + `DivIL` (random masking) | $71.27 \pm 1.36$ | $69.25 \pm 0.81$ |

### C.2  Interpretation Visualization of Over-invariance on Graph

To demonstrate the over-invariance issue on the real datasets, we use the interpretable GNN architecture (XGNN) to extract $G_c$, it brings an additional demonstration that over-invariance risk occurs in the real-graph datasets, which may facilitate human understanding in practice.

We provide some visualizations of invariant subgraphs learned by Chen et al. (2022). First, we illustrate the examples in SPMotif under the biases of 0.6 and 0.9 shown in Figure 8 and Figure 9. We use pink to color the ground truth nodes in $G_c$, and denote the relative attention strength with edge color intensities. We also

provide some interpretation visualization examples in DrugOOD datasets shown in Figure 10 to Figure 11. We use the edge color intensities to denote the attention of models that pay to the corresponding edge.

We have two observations based on the above visualization examples. 1) Invariant learning methods tend to extract the subgraph of the ground truth invariant subgraph(Figure 8, 9, 10, 11), thus verifying the over-invariance issue. 2) When confronted with multiple invariant subgraphs (Figure 10, 11), these learning methods usually focus on only one of them rather than considering all the invariant subgraphs. In the fields of computer vision and natural language processing, we do not visualize them here due to the ambiguous definitions and interpretations of invariance and spurious effects. The above two observations aligned with our synthetics experiments in Figure 3 measured by the strength.

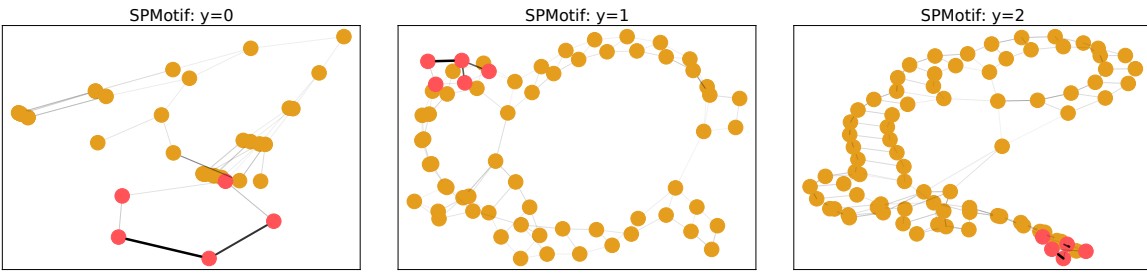

Figure 8: Over-invariance visualizations of examples from SPMotif-Struc under bias= 0.6 (Chen et al. (2022)).

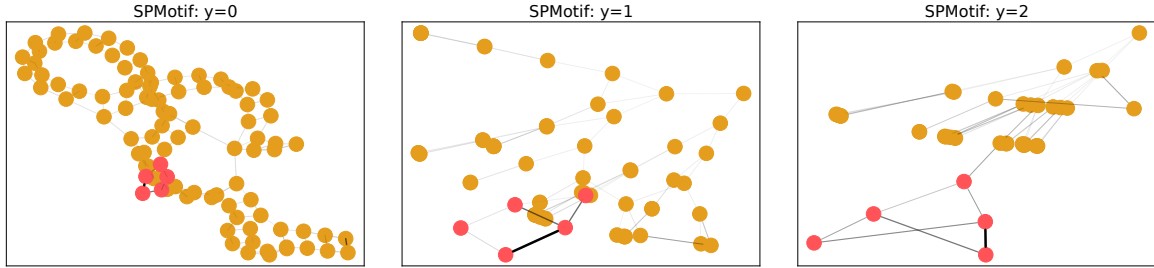

Figure 9: Over-invariance visualizations of examples from SPMotif-Struc under bias= 0.9 (Chen et al. (2022)).

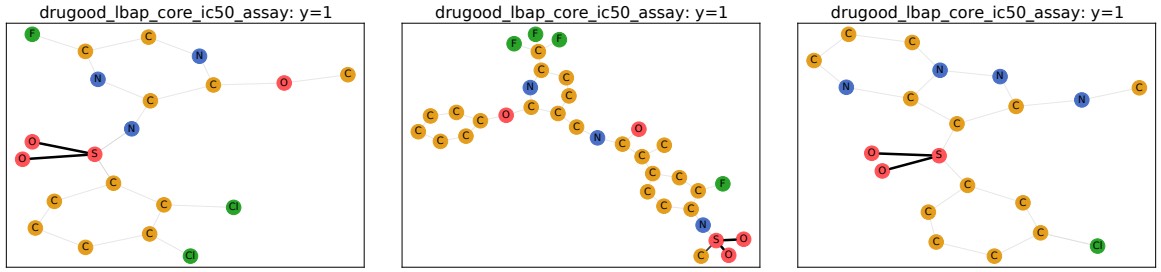

Figure 10: Over-invariance visualizations of activate examples ($y = 1$) from DrugOOD-Assay (Chen et al. (2022)).

# D  Software and Hardware

We implement our methods with PyTorch (Paszke et al., 2019) and PyTorch Geometric (Fey & Lenssen, 2019). We ran our experiments on Linux Servers installed with 3090 graphics cards and CUDA 10.2.

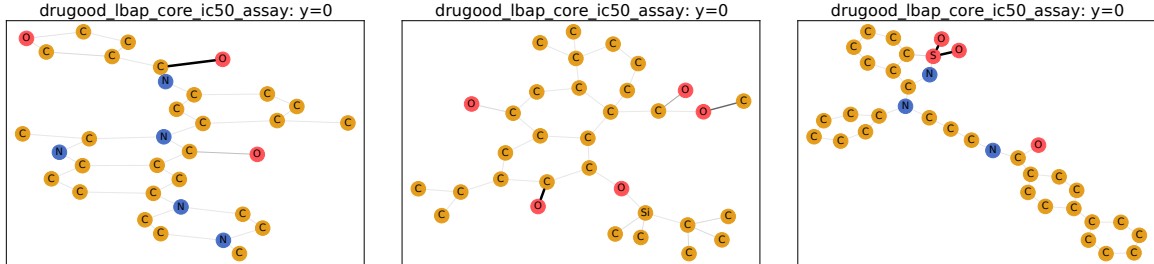

Figure 11: Over-invariance visualizations of inactivate examples ($y = 0$) from DrugOOD-Assay (Chen et al. (2022)).

