# OpenReview forum: "DivIL: Unveiling and Addressing Over-Invariance for Out-of- Distribution Generalization"
_TMLR — Accepted by TMLR_

### Review · Reviewer_VXRv · 2024-11-11

**Summary Of Contributions:**

In this work, the authors offer novel insights into the well-studied out-of-distribution generalization problem from the standpoint of invariant learning (IL), whose core algorithm (IRM) adopts the assumption that *spurious* features (a) harm out-of-distribution generalization and (b) are inconsistent across environments (also sometimes referred to in this literature as groups). Several theoretical and empirical challenges to this idea have entered the literature since its introduction in 2019, to which the authors add another --  over-regularization in the feature space may lead to a loss of important details, an effect which they call *over-invariance*. The authors claim that over-invariance degrades generalization performance, propose Diverse Invariant Learning (DivIL), combining both invariant constraints and unsupervised contrastive learning with randomly masking mechanism to promote richer and more diverse invariance, and offer experiments on 12 benchmarks to validate the efficacy of their method.

**Audience:**

Yes

**Broader Impact Concerns:**

I see no broader impact concerns.

**Claims And Evidence:**

Yes

**Requested Changes:**

1. You write "known as the in-distribution assumption (IID assumption) (Liu et al., 2023; Peters et al., 2016c)". I.I.D. stands for independent and identically distributed, per your citations and general use. Please modify this sentence to reflect this.
2. Please add summaries of the important findings contained in each table to the captions provided for that table.
3. Table 1 should be relocated to the experiments section.
4. Your most robust experiments are on graphs; therefore, those results should be presented first.
5. Please scope your third claim in the experiments section with greater precision. "can attain state-of-the-art performance"; from my reading of the paper, it appears that your method can claim state of the art only on graphs, and that even there, the differences are not statistically significant.
6. The claim that the authors are the first to discover what they call *over-invariance* is overly broad and should be reworded. Similar phenomena have been extensively studied and documented under various names, such as *underfitting* (Goodfellow, Ian, et al. Deep Learning. MIT Press, 2016).
7. The description of DivIL mentions at least three new mechanisms introduced that break with prior work; **Random Data Augmentation**, **Unsupervised Contrastive Learning** and **Random Masking**. The authors should provide ablations on the impact of each of all changes in isolation on at least some of their datasets (ideally in the appendix) and reference them in the main paper.

**Strengths And Weaknesses:**

The authors provide a very useful introduction section which helps bring readers up to speed on the subfield. Their figure 1 illustration is well designed and illuminating. Experiments which show how much benefit straightforward pragmatic interventions can offer in a particular setting will always be of interest to at least some TMLR readers. The authors' experiments on graphs show consistent (if not statistically significant) improvement over the strongest baseline method.

Please see the requested changes section for weaknesses.

---

> ### Author Response · Authors · 2024-12-02
>
> We thank reviewer VXRv for your insightful comments.
>
> ---
>
> **Question 1:** The claim that the authors are the first to discover what they call over-invariance is overly broad and should be reworded. Similar phenomena have been extensively studied and documented under various names, such as underfitting.
>
> **Answer 1:** Thanks for your thoughtful feedback. We firmly hold the view that there are two major differences between underfitting and over-invariance:
>
> 1.	**Different Triggers:** As we discussed in Sec. 3.2, underfitting is caused by insufficient training, while the over-invariance issue is caused by **the limited environmental diversity and the over-regularization via the representation space**.
>
> 2.	**Different Applicable Scenarios:** Underfitting is for a general deep learning problem while over-invariance is specified on the **out-of-distribution generalization** solved by the invariant learnings.
>
> Therefore, we sincerely believe there are some misunderstandings in our work, and the causes and scenarios of the over-invariance phenomena we raise are quite different.
>
> **Question 2:** The description of DivIL mentions at least three new mechanisms introduced that break with prior work; Random Data Augmentation, Unsupervised Contrastive Learning, and Random Masking. The authors should provide ablations on the impact of each of all changes in isolation on at least some of their datasets and reference them in the main paper.
>
> **Answer 2:** Thank you for your valuable advice. We agree that ablation experiments are important, however, we believe there may be some misunderstandings here. In fact, we **have conducted comprehensive ablations** to evaluate different mechanisms of DivIL in our work. Below we will analyze these ablation experiments in detail:
>
> 1.	Figure 6 shows the impact of **data augmentations**, such as edge removal, node dropping, and subgraph extraction in the graph;
>
> 2.	Table 4 shows the comparisons with and without **unsupervised contrastive learning** and Figures 7(b) and 7(c) explore the weight sensitivity of UCL,
>
> 3.	Figure 7(a) shows the ablation study on the different **random masking** percentage.
>
> In addition to the previous ablations, we further revised the paper to include analyses on the difference between random masking and dropout, and interpretation visualizations of real-world datasets in Appendix C. We believe that these additional ablations provide a comprehensive overview. If you require further information, please feel free to let us know.
>
> **Question 3:** “known as the in-distribution assumption (IID assumption)”. I.I.D. stands for independent and identically distributed, per your citations and general use. Please modify this sentence to reflect this.
>
> **Answer 3:** Thank you for your constructive advice. We have revised it to the sentences as follow:
> > “These methods heavily rely on the assumption that training and testing data come from the independent and identically distribution(I.I.D assumption).”
>
> **Question 4:** Some suggestions for improving the presentation of the paper.
>
> **Answer 4:** Thanks for your detailed feedback. We have followed your suggestion to fix the imprecise expressions one by one:
>
> -	Add the findings in the table caption highlighted in orange.
>
> -	Relocate Table 1 to the Experiments section.
>
> -	Reorder our experiments to present the graph exp. first.
>
> -	Revised the third claim in the experiments to “Experiments conducted on 12 benchmarks, 4 different invariant learning methods across 3 modalities (graphs, vision, and natural language) demonstrate that DivIL effectively enhances the out-of-distribution generalization performance, verifying the over-invariance insight.”
>
> ---
>
> Thanks for your insightful questions. Hope our explanations on the over-invariance issue and the ablation study could address your concerns. Please let us know if there is more to clarify.

---

### Review · Reviewer_wSGc · 2024-11-15

**Summary Of Contributions:**

This paper aims to improve the state-of-the-art of out of distribution detection by improving invariant learning techniques. Invariant learning aims to learn features without spurious correlations, allowing classifiers to generalize to new environments. Previous works potentially suffer from over-invariance, where a subset of invariant features are discarded unnecessarily, degrading generalization. The authors build off of the previous theoretical definition of invariance to rigorously define invariant features, over-invariance, and over-invariant features. The feature norm under data masking is empirically investigated to try to identify over-invariance in practice. DivIL is proposed as a solution to over-invariance, adding contrastive learning and feature masking to existing methods. The experiments benchmark invariant learning methods on multiple datasets across multiple modalities. The proposed method shows similar or increased performance in each evaluation.

**Audience:**

Yes

**Broader Impact Concerns:**

I do not have any concerns about potential broader impacts of this paper.

**Claims And Evidence:**

Yes

**Requested Changes:**

C1: Complete the README.md in CIGA_NC_latest.

C2: I think “exceeded human-level performance” in the first sentence of the introduction is overstating previous results.

C3: Definition 3.2 does not appear to be grammatically correct. Suggested rewording: “Suppose selection function $I = \\{0, 1\\}^k$, such that the subset $Z^c = Z ◦ I$ of the latent feature $Z ∈ R^k$ is an invariant feature of label $Y$ under both train distribution $P_{tr}$ and test distribution $P_{te}$. Therefore $Z^c$ satisfies”

C4: Remark 4.1 cites Equation 6 when it should be 7.

**Strengths And Weaknesses:**

S1: The provided code is well documented. The use of numerous README files makes it easy to understand how the code can be used. The appendix further expands on experiment details.

S2: The theoretical definitions laid out in Definitions 3.2 and 3.3 builds off of previous work (Definition 3.1) to provide new insights into the structure of invariant features and the issue of over-invariance.

S3: The extensive experiments benchmark multiple existing methods and the proposed method on multiple datasets across image, natural language, and graph modalities. The proposed method shows similar or increased performance in each evaluation.


W1: Section 3.3 does not clearly follow from Section 3.2. Definition 3.3 states that there should not be a *feature subset* in $Z^c$ that changes the expected *class prediction*. Definition 3.5 instead values *embedding norms* under *masked input values*, where the masked values are also valid samples from the distribution. Why not use the decomposed logits, as in the equation above Theorem 3 in Invariance Principle Meets Information Bottleneck for Out-of-Distribution Generalization (Ahuja et al. 2021), to directly examine changes in class prediction?

W2: How does Figure 3 relate to Remarks 3.6 and 4.1? Doesn’t Figure 3 show that the strength is nonzero without DivIL? Is averaging across values hiding the zeros? How would adding DivIL to the methods change the results in Figure 3?

W3: Why should the proposed random masking method alleviate the over-invariance problem more than a more common method like dropout? The lack of an ablation further obscures the impact of this choice.

W4: Are the results reported in the tables statistically significant? If so, under what test of significance? The lack of a large margin between some results makes this ambiguous.

W5: Is there a paper that should be cited for the particular unsupervised contrastive learning loss function used? “Representation learning with contrastive predictive coding” (Oord et al., 2019) can be credited for the general formulation, but I’m not familiar with any uses of the negative squared norm in previous literature.

---

> ### Author Response · Authors · 2024-12-02
>
> We thank reviewer wSGc for your attentive reading and questions.
>
> ---
>
> **Question 1:** Section 3.3 does not clearly follow from section 3.2. Why do not use decomposed logits to directly examine changes in class prediction in the synthetic dataset.
>
> **Answer 1:** Thank you for your kind feedback. The overall logic between those two sections is that, in Section 3.2, we delve into two potential factors contributing to the challenge of attaining invariant features: (1) limited environmental diversity and (2) over-regularization via the representation space, ultimately leading to the over-invariance issue. In Section 3.3, we validate the occurrence of the over-invariance issue through the synthetic experiments.
>
> In Section 3.3, Definition 3.5 introduced a metric strength to quantify the preferences of invariant learning among various components of invariant features, distinct from the aim of assessing changes in class prediction through decomposed logits. The essence of the strength metric is this: As its value decreases, the model's attention towards those features diminishes, indicating a lower level of focus. While the decomposed logits are utilized for mathematical theorem proof, they do not possess these attributes.
>
> **Question 2:** How does Figure 3 relate to Remarks 3.6 and 4.1? Doesn’t Figure 3 show that the strength is nonzero without DivIL? Is averaging across values hiding the zeros? How would adding DivIL to the methods change the results in Figure 3?
>
> **Answer 2:** Thank you for your thoughtful feedback. First, we would like to reiterate what Figure 3 is describing. Specifically, Figure 3 illustrates the synthetic results that a subset of the invariant features has lower strength than others; thus, aligned with the key concept of this work: the over-invariance issue in Definition 3.3.
>
> Remark 3.6 serves as a structured recapitulation of the insights presented in Figure 3 to show the presence of a subset of the invariant feature whose strength may tend to zero in the extreme case. We have revised the zero strength to the comparison between different subsets within the invariant features, which is a loose conclusion and more directly connected to Figure 3.
>
> Remark 4.1 captures the core concept depicted in Figure 5, distinct from Figure 3, showcasing the heightened strength upon integrating unsupervised contrastive learning into IRM and VREx. This augmentation highlights DivIL's capacity to enhance model attention towards the invariant features by introducing diversity and random.
>
> **Question 3:** Ablations on random masking and dropout.
>
> **Answer 3:** Thank you for your constructive advice.
> We have added the experiments based on 3 invariant learning methods in computer vision task to show the effectiveness of random masking (more details seen in Appendix C.1). The results in Table 1 showed that random masking outperforms dropout in the test set.
>
> Table 1. Ablations on random masking and dropout in CMINIST.
> |             |   train      |   test       |
> |-------------|--------------|--------------|
> | IRM+rm      | 70.93 ± 0.29 | **66.40 ± 1.39** |
> | IRM+dropout | 70.65 ± 0.54 | 66.19 ± 1.78 |
> | VREX+rm     | 72.67 ± 0.93 |**67.50 ± 1.45** |
> | VREX+dropout| 72.54 ± 1.72 | 66.73 ± 1.34 |
> | Fishr+rm    | 71.27 ± 1.36 | **69.25 ± 0.81** |
> | Fishr+dropout| 71.36 ± 1.41| 69.20 ± 0.77 |
>
>
> **Question 4:** lack of statistically significant in the table.
>
> **Answer 4:** Thank you for your kind inquiry. We firmly believe that there is a misunderstanding. In fact, almost all tables in our work include the standard deviation, which already assesses the statistical significance of the random seeds.
>
> **Question 5:** uses of the negative squared norm in the paper.
>
> **Answer 5:** Thank you for your attentive reading. We are sorry for the mistake in this equation, there should be no negative signs before the squared norm aligned with the general form in the paper [1], and we have revised them in this work. Additionally, we have included [1] in our work to enrich the discussion on unsupervised contrastive learning methods. Your understanding is greatly appreciated.
>
> **Question 6:** Detailed suggestions for article improvement  in the first sentence of the introduction is overstating previous results.
>
> **Answer 6:** Thank you for your detailed feedback. We will correct these details point by point:
>
> -	Change “exceeded human-level performance” to “achieve great success” to avoid overstating previous results.
>
> -	Correct the grammar of the Definition 3.2 (highlighted in orange).
>
> -	Correct the equation reference from Eq. 6 to Eq. 7.
>
> -	Complete the README.md.
>
> ---
>
> Thank you again for your comments. Hope our explanation of the definitions and remarks, and the experiment on dropout could address your concerns.
>
> [1] Representation learning with contrastive predictive coding. Oord et al. Arxiv 2018.

---

> > ### Comment · Reviewer_wSGc · 2024-12-02
> >
> > Overall I believe that these revisions addressed my concerns.
> >
> > The one exception is Question 4. For a high level understanding of the issue with the current implementation, I recommend reading [Creating Confidence Intervals for Machine Learning Classifiers](https://sebastianraschka.com/blog/2022/confidence-intervals-for-ml.html). A more rigorous overview is provided in [Model Evaluation, Model Selection, and Algorithm
> > Selection in Machine Learning](https://arxiv.org/pdf/1811.12808).
> >
> > In short, error bars only guarantee statistical significance if they do not overlap at the desired significance level (e.g. 1.96 standard deviations for 95% confidence). This is clearly not the case for many of the results reported here. For example, DivIL and CNCP on EC50-Assay in Table 1 have a standard deviation difference of only [~1](https://www.wolframalpha.com/input?i=%2877.00+-+73.74%29%2F%281.52%5E2+%2B+2.62%5E2%29%5E%281%2F2%29), less than the 1.96 required for statistical significance.
> >
> > I think that this lack of statistical significance should be clearly communicated, but further changes are not needed.

---

> > > ### Author Response · Authors · 2024-12-03
> > >
> > > Thank you for your thoughtful feedback and for acknowledging our efforts. We are glad our responses addressed your concerns and appreciate your valuable advice.

---

### Review · Reviewer_uhmW · 2024-11-20

**Summary Of Contributions:**

The paper investigates the phenomenon of over-invariance in invariant learning methods, where overly restrictive invariant constraints lead to the loss of discriminative features and hurt the OOD generalization. The authors propose a new method, DivIL, to address this limitation. DivIL employs a contrastive loss, where the positive pair of a sample is obtained by random dropping a subset of it (e.g., masking a subset of pixels to 0 for images). DivIL is tested across 12 datasets spanning computer vision, natural language, and graph learning tasks, and shows improvements over existing invariant learning methods.

**Audience:**

Yes

**Broader Impact Concerns:**

No concern.

**Claims And Evidence:**

No

**Requested Changes:**

- Explain in details how the proposed method extends beyond existing work in invariant learning and contrastive learning.
- Revise the theoretical framework to reduce reliance on overly simplistic assumptions (e.g., Gaussian feature distributions, clearly separable spurious and invariant features). Or discuss how the proposed framework fits real-world data.

**Strengths And Weaknesses:**

Strengths:
- Evaluation is comprehensive.
- Besides images, the methods generalize to data in different domain, such as graph or natural language.
- The idea of over-invariance is interesting.

Weaknesses:
- The addition of the contrastive term does not substantially advance the theoretical or practical understanding of invariant learning. There are existing works that already explore this area [1] [2]. While the proposed method uses a different augmentation strategy (similar to dropout), it feels incremental rather than a significant leap forward.
- While the paper discusses an interesting idea of over-invariance, it relies heavily on synthetic experiments to validate its claims. This does not convincingly demonstrate the real-world impact of over-invariance.
- Features are modeled as Gaussian distributions with clear separability between invariant and spurious components. Such assumptions may not hold in real-world data, limiting the applicability of the theoretical insights.

---

> ### Comment · Reviewer_wSGc · 2024-11-22
>
> For the first weakness, do you believe that the paper is so incremental that it would not be interesting to some individuals in TMLR's audience? I think that this paper satisfies that part of TMLR's evaluation criteria.

---

> ### Author Response · Authors · 2024-12-02
>
> We thank reviewer uhmW for your valuable comments.
>
> ---
>
> **Question 1:** The limit of showing the over-invariance just on the synthetic Gaussian distribution datasets, and lacks demonstration on the real-world datasets. This does not convincingly demonstrate the real-world impact of over-invariance.
>
> **Answer 1:** Thank you for your constructive advice. We add discussion about the over-invariance in the real-world datasets in Section 3 and provide additional visualization examples in Appendix C.2. We utilize an interpretable Graph Neural Network (GNN) to extract invariant subgraphs from datasets displaying strong spurious correlations, specifically spmotif-0.6 and spmotif-0.9, as well as from the real-world AI-aided drug discovery dataset, Drugood-Assay. In our analysis, we highlight the ground truth nodes in $G_c$ with a pink hue, and denote the relative attention strength with edge color intensities following [3]. We have two observations: (1) Invariant learning methods tend to extract the subgraph of the ground truth invariant subgraph. (2) When confronted with multiple invariant subgraphs, these learning methods usually focus on only one of them rather than considering all the invariant subgraphs.
>
> The above two observations aligned with our synthetics experiments in Figure 3 and thus verify our proposed over-invariance issue in invariant learning.
>
>
> **Question 2:** Features in the synthetic dataset are modeled as Gaussian distributions with clear separability between invariant and spurious components. Such assumptions may not hold in real-world data, limiting the applicability of the theoretical insights.
>
> **Answer 2:** Thank you for your question. The intentionally crafted synthetic Gaussian dataset, featuring a clear distinction between invariant and spurious components, serves the purpose of simplifying the intricate real dataset by utilizing controllable hyperparameters. Our aim is to distinguish between these components and effectively measure their “strength” learned by different invariant learning methodologies, particularly within the invariant features exhibiting varying variances amidst changes in the diverse spurious features, as illustrated in Figures 3 and 5. To expand on our understanding of over-invariance, as demonstrated in Answer 1, we provide additional interpretive visualizations to highlight and confirm the presence of this phenomenon.
>
> **Question 3:** Explain in details how the proposed method extends beyond existing work in invariant learning and contrastive learning.
>
> **Answer 3:** Thank you for your attentive reading and question.
> 1) As you previously mentioned, our approach, DivIL, exhibits a “generalization ability to data in different domains, like graph, cv, and nlp”. This extension marks a significant advancement over prior invariant learning typically confined to individual domains, such as computer vision or graphs.
> 2)  Moreover, DivIL stands out for its flexibility and plug-and-play adaptability to a range of invariant learning techniques, including IRM, VREx, CIGA, and Fishr, all of which we have experimentally evaluated in our study.
> 3) Additionally, a key contribution of our paper is the introduction and formal definition of the over-invariance issue, as you noted “interesting idea”, in traditional invariant learning methods accompanied by a thorough theoretical analysis. Our DivIL is designed to validate this insight through a contrasting lens by integrating random data augmentation, unsupervised contrastive learning and random mask into the invariant learning process, distinguishing itself from both contrastive and invariant methods.
>
> **Question 4:** Contrastive term feels incremental rather than a significant leap forward. Existing works [1] [2] already explore this area.
>
> **Answer  4:** Thanks for your feedback. We apologize for the confusion as the references to papers [1] and [2] were not provided, making it challenging for us to fully grasp your question. Our work diverges from assessing the efficacy of contrastive terms; instead, we concentrate on uncovering the over-invariance concern across different invariant learning methods. Inspired by this, we designed the DivIL, which introduces three specific mechanisms. It is over-invariance-oriented random data augmentation and random masking, leveraged by unsupervised contrastive learning, that effectively enhances the diversity and further relieves the over-invariance. We would be more than happy to discuss these further.
>
> ---
>
> Thank you again for sharing your thoughts with us. Hope our real-dataset visualization observations and explanations of the over-invariance could address your concerns. We are happy to address your further questions!
>
> [3] Learning causally invariant representations for out-of-distribution generalization on graphs. Chen et al. NeurIPS 2022.

---

### Decision · Action_Editor_p9ac · 2025-01-24

**Recommendation:** Accept as is

**Comment:**

This paper was reviewed by three experts who provided valuable comments and insightful suggestions. After constructive discussion, two reviewers agreed to recommend the paper for acceptance to TMLR. Following a thorough review of the revised version and further discussion, I also recommend acceptance, as all reviewers concur that the experiments support the main claim regarding the over-invariance issue in real-world problems.

Moreover, please ensure the discussion with Reviewer uhmW is included in the camera-ready version, specifically: “Explain in detail how the proposed method extends beyond existing work in invariant learning and contrastive learning.”

**Audience:**

Researchers focusing on model generalization and improving model performance in out-of-distribution scenarios would find this work valuable.

**Claims And Evidence:**

This paper explores invariant learning (IL) for out-of-distribution generalization, highlighting that strong IL can result in the loss of important details. Experiments demonstrate that various IL methods can cause this issue. To address it, we propose Diverse Invariant Learning (DivIL), a simple approach that combines unsupervised contrastive learning and a random masking mechanism to complement the invariant constraints. DivIL is adaptable to a range of IL methods and effectively mitigates this problem.